# Anti-asthmatic effect of nitric oxide metallo-donor FOR811A [*cis*-[Ru(bpy)$_2$(2-MIM)(NO)](PF$_6$)$_3$] in the respiratory mechanics of Swiss mice

Paula Priscila Correia Costa[1,2]*, Stefanie Bressan Waller[1]*, Gilvan Ribeiro dos Santos[3], Fladimir de Lima Gondim[3], Daniel Silveira Serra[3], Francisco Sales Ávila Cavalcante[3], Florêncio Sousa Gouveia Júnior[4], Valdir Ferreira de Paula Júnior[5], Eduardo Henrique Silva Sousa[4], Luiz Gonzaga de França Lopes[4], Wesley Lyeverton Correia Ribeiro[2]*, Helena Serra Azul Monteiro[2]

**1** Department of Veterinary Clinics, Faculty of Veterinary, Federal University of Pelotas, Pelotas, RS, Brazil, **2** Department of Physiology and Pharmacology, Faculty of Medicine, Federal University of Ceará, Fortaleza, CE, Brazil, **3** Laboratory of Biophysics of Respiration, Superior Institute of Biomedical Sciences, State University of Ceará, Fortaleza, CE, Brazil, **4** Laboratory of Bioinorganic, Department of Organic and Inorganic Chemistry, Federal University of Ceará, Fortaleza, CE, Brazil, **5** Laboratory of Biotechnology and Molecular Biology, Superior Institute of Biomedical Sciences, State University of Ceará, Fortaleza, CE, Brazil

* paulapriscilamv@yahoo.com.br (PPCC); waller.stefanie@yahoo.com.br (SBW); wesleylyeverton@yahoo.com.br (WLCR)

**Data Availability Statement:** All relevant data are within the paper and its Supporting Information files.

## Abstract

We aimed at evaluating the anti-asthmatic effect of *cis*-[Ru(bpy)$_2$(2-MIM)(NO)](PF$_6$)$_3$ (FOR811A), a nitrosyl-ruthenium compound, in a murine model of allergic asthma. The anti-asthmatic effects were analyzed by measuring the mechanical lung and morphometrical parameters in female Swiss mice allocated in the following groups: untreated control (Ctl +Sal) and control treated with FOR811A (Ctl+FOR), along asthmatic groups untreated (Ast +Sal) and treated with FOR811A (Ast+FOR). The drug-protein interaction was evaluated by *in-silico* assay using molecular docking. The results showed that the use of FOR811A in experimental asthma (Ast+FOR) decreased the pressure-volume curve, hysteresis, tissue elastance, tissue resistance, and airway resistance, similar to the control groups (Ctl+Sal; Ctl+FOR). However, it differed from the untreated asthmatic group (Ast+Sal, $p<0.05$), indicating that FOR811A corrected the lung parenchyma and relaxed the smooth muscles of the bronchi. Similar to control groups (Ctl+Sal; Ctl+FOR), FOR811A increased the inspiratory capacity and static compliance in asthmatic animals (Ast+Sal, $p<0.05$), showing that this metallodrug improved the capacity of inspiration during asthma. The morphometric parameters showed that FOR811A decreased the alveolar collapse and kept the broncho-constriction during asthma. Beyond that, the molecular docking using FOR811A showed a strong interaction in the distal portion of the heme group of the soluble guanylate cyclase, particularly with cysteine residue (Cys141). In summary, FOR811A relaxed bronchial smooth muscles and improved respiratory mechanics during asthma, providing a protective effect and promising use for the development of an anti-asthmatic drug.

**Funding:** This study was financed by the Brazilian Institutes Coordenação de Aperfeiçoamento de Pessoal de Nível Superior (CAPES) under Grant CAPES/FUNCAP - 88881.166839/2018-01 (L.G.F. Lopes); and Conselho Nacional de Desenvolvimento Científico e Tecnológico (CNPq) under Grant CNPq - 303355/2018-2 (L.G.F. Lopes) and 308383/2018-4 (E.H.S. Sousa).

**Competing interests:** The authors have declared that no competing interests exist.

## Introduction

Asthma is a respiratory condition characterized by intense inflammation and hyperreactivity of the airways, leading to a significant increase in mucus secretion and respiratory resistance [1]. Therefore, granulocytes–eosinophils, lymphocytes, macrophages, and mast cells–are activated triggering contraction of the airway smooth muscle, in addition to microvascular leakage and mucus secretion [2, 3]. Thus, the main signs of asthma are shortness of breath, wheezing, coughing and tightness in the chest [4], which can even lead to death, mainly in individuals aged 65 and over [5].

According to the Global Initiative for Asthma [6], the current treatments consist of preventing airflow limitation and improving pulmonary conditions. Oral corticosteroids, ß-agonists, anti-inflammatories, and inhaled bronchodilators [7, 8] are examples of drugs available to control the disease. However, severe cases [9] are more difficult to control because said drugs may not be responsive, even at high doses [6, 10], which justifies the need for effective therapeutic alternatives.

One of the strategies for regulating the pathophysiological processes of asthma is the use of nitric oxide (NO), a small molecule of great importance in the modulation of cardiovascular, immunological, nervous, and even pulmonary processes [11]. In the respiratory tract, the nose and paranasal sinuses are the main production sites of exhaled NO [12], which activates the soluble guanylate cyclase (sGC) to significantly increase the production of cyclic guanosine monophosphate (cGMP).

In turn, the high production of cGMP will activate important protein kinases, which will act in the regulatory functions, such as smooth muscle relaxation, neuronal transmission, inhibition of platelet aggregation and regulation of vascular/bronchial tone and protective function against excessive bronchoconstriction [11, 12]. Having said that, the conditions of bronchoconstriction are modelled by NO, that acts as bronchodilator in the human lungs [13]. However, NO of endogenous origins has a short half-life, and deficiencies in its production that result in its inactivation, generating pathological conditions [11].

The use of compounds able to modulate the NO/sGC/cGMP pathways are valuable therapeutic agents, with emphasis on organic-based molecules complexed with ruthenium. Studies in the murine model have been promising with these potential metallodrugs as NO donors, such as $cis$-[Ru(bpy)$_2$(ImN)(NO)]$^{3+}$, where bpy = 2,2'-bipyridine and ImN = imidazole– known as FOR0811 –for the control of arterial hypertension [14]. Moreover, the nitrosyl-ruthenium complexes, such as [Ru(terpy)(bdq)NO$^+$]$^{3+}$–known as TERPY–, are promising due to the relaxant effect on the smooth muscles of the airways [15] and in the control of asthma [16]. Another nitrosyl-ruthenium complex is $cis$-[Ru(bpy)$_2$(2-MIM)(NO)](PF$_6$)$_3$ –known as FOR811A–, whose effect on acute inflammation [17] has been promising, however, it had never been studied for allergic asthma.

Given the potential of organic-based compounds of the nitrosyl-ruthenium, this study aimed at evaluating, for the first time, the effect of $cis$-[Ru(bpy)$_2$(2-MIM)(NO)](PF$_6$)$_3$ – FOR811A –on the mechanics in murine model induced to allergic asthma.

## Materials and methods

### Synthesis of ruthenium complexes

The ruthenium compound complexed with the nitric oxide (NO) molecule–$cis$-[Ru(bpy)$_2$(2-MIM)(NO)](PF$_6$)$_3$ –was synthesized at the Laboratory of Bioinorganic (Federal University of Ceará, Fortaleza/CE, Brazil), according to Gouveia Júnior et al. [18]. This compound was called FOR811A, and its flat structure is shown in Fig 1. Briefly, 0.4 mmol of the precursor

**Fig 1. Chemical structure of the ruthenium complex cis-[Ru(bpy)$_2$(2-MIM)(NO)](PF$_6$)$_3$.**

cis-[Ru(bpy)$_2$Cl$_2$] was reacted with equimolar amount of 2-methylimidazole in ethanol under reflux and magnetic stirring. After 2 hours, an equimolar amount of NaNO$_2$ dissolved in water was added, keeping the system in the previous conditions for 2 more hours. After that, the solvent was removed through rotary evaporation and the crude product was mixed with a 10% HPF$_6$ solution, giving the desired product as a light orange solid. Yield: 45%. HRESI-MS (+): [M– 3PF$_6$]$^{2+}$ theoretical: 263,0459 (C$_{24}$H$_{22}$N$_7$ORu$^{2+}$); experimental: 263,0459. Elemental analysis: Theoretical (C$_{24}$H$_{22}$F$_{18}$N$_7$OP$_3$Ru): C, 30,01; H, 2,31; N, 10,21%. Experimental: C, 30,06; H, 2,38; N, 10,35%.

## Experimental animals

For the experimental study, 15-weeks-old female Swiss mice (*Mus musculus*, n = 40) weighing between 25 and 30 grams were used. The animals were obtained from the Central Bioterium (Federal University of Ceará, Fortaleza/CE, Brazil) and kept in cages containing five animals each. They were housed in controlled conditions of humidity, temperature (22°C), 12/12h light-dark cycle, and commercial diet and water *ad libitum*. This murine model was chosen because it does not show consanguinity, similarly to humans [19, 20], and because female mice seem to be more sensitive to develop allergic inflammations [21, 22]. All procedures were approved by the Ethics Committee for Use in Animals (*Comitê de Ética para Uso em Animais–CEUA*, State University of Ceará, no. 2068307/2018).

## Experimental design and treatments

The animals were allocated to four experimental groups (n = 10 each): untreated control receiving saline solution (Ctl+Sal); control treated with FOR811A (Ctl+FOR); untreated

asthmatic receiving saline solution (Ast+Sal); asthmatic treated with FOR811A (Ast+FOR). On days 0, 7, and 14 of the experiment, the animals were sensitized (intraperitoneal injection, 0.2 mL) with 0.9% of saline solution (NaCl) for control groups (Ctl+Sal; Ctl+FOR) or ovalbumin (Sigma-Aldrich®, St. Louis/MO, USA; 100 μg, dissolved in 5 mg of aluminum hydroxide–AlOH) for asthmatic groups (Ast+Sal; Ast+FOR).

On days 26, 27, and 28, all animals were placed individually in an acrylic box ($30 \times 15 \times 20$ cm) coupled to an ultrasonic nebulizer (US-1000, ICEL, São Paulo, Brazil) and submitted to the inhalation challenge. Control groups (Ctl+Sal; Ctl+FOR) were challenged to inhale 0.9% NaCl for 20 minutes, whereas the asthmatics groups (Ast+Sal; Ast+FOR) were challenged to inhale 50 μg ovalbumin at a concentration of 10 mg/mL for 20 minutes. All animals also received tramadol (5 mg/kg/8h, Cronidor 2%®, Agener União Saúde Animal Ltda., São Paulo/SP, Brazil) as an analgesic method to minimize pain, suffering and distress. Additionally, the post-challenge care included the monitoring the animals twice a day by veterinarians (8:00 am and 5:00 pm) for possible behavioral changes through the application of the Grimmace Scale. The researchers were previously trained to apply the "humanitarian endpoint", according to Brazilian legislation, if any animal had its welfare compromised. Normal interaction with other animals, amount of feeding, and volume of water ingested were also monitored.

On day 29, all animals received a single dose of oral treatment by gavage (0.2 mL). Control groups (Ctl+Sal; Ctl+FOR) received 0.9% NaCl, whereas the asthmatics groups (Ast+Sal; Ast +FOR) were treated with FOR811A at the concentration of 0.75 mg/kg. Finally, on day 30, the association between 10% ketamine hydrochloride (300 mg/kg, Cetamin®, Syntec, São Paulo/SP, Brazil) and α2-adrenergic receptor agonists 2% xylazine hydrochloride (30 mg/kg—Sedanew®, Vetnil, São Paulo/SP, Brazil) was used for the euthanasia of the mice by anesthetic overdose.

## Evaluation of the mechanical lung measurements

An integrated platform was used for collecting data on pulmonary mechanics measurements (S1 Fig). The use of the mechanical respirator for small animals (FlexiVent, SCIREQ, Montréal, Canada) made it possible to apply arbitrary waveforms to the injected volumes or pressures applied to the lung, with the simultaneous acquisition of all determinant variables in the organ mechanics. Besides, the forced oscillation technique associated with the constant phase model [23] was used to extract the respective quantities, as it performs well in separating the mechanical properties of the airways and lung tissue.

To obtain the pressure-volume curve points, the pressure in the trachea was raised to 30 cmH$_2$O at pre-established equally spaced pressure intervals, with records of the plateau volume values corresponding to these pressures. The same procedure, with decreases in pressure, was performed to obtain the expiratory branch of the curve. Thus, it was possible to simultaneously obtain the measurement of static compliance ($C_{stat}$), estimation of inspiratory capacity (IC), and calculation of the curve area. For the measurements of the parameters of the constant phase model, a quick-prime disturbance was applied, which consisted in the imposition of airflow with an amplitude corresponding to the sum of sine waves of frequencies between 1.00 to 20.5 Hz. The pressure and flow obtained from this disturbance were used to calculate the impedance of the respiratory system (Zrs), which was adjusted to the constant phase model [24]. Therefore, the values of airway resistance, also known as Newtonian resistance ($R_N$), and the tissue resistance (G), tissue elastance (H) and hysteresis (η) were obtained.

After the connection of the animal to the ventilator, it was paralyzed with pancuronium bromide (0.5 mg/kg). During the five minutes that elapsed, possible leaks, obstructions, as well

as the corrections in the positioning of the animal's body to the ventilator and the confirmation of the absence of spontaneous inspirations were verified. Therefore, 12 quick-prime perturbations were performed to determine the parameters of the constant phase model ($R_N$, G, H, and η).

## Morphometrical parameters

After the pulmonary mechanics measurements, the lungs were perfused with saline, removed *en bloc*, kept at functional residual capacity and fixed in Millonig's formaldehyde (100 mL HcHO, 900 mL $H_2O$, 18.6 g $NaH_2PO_4$, 4.2 g NaOH) [25]. The pulmonary section slides were stained with Hematoxylin-Eosin (HE) and were examined under optical microscopy by blind reading, in which the experimental groups were unknown during the evaluator's reading. For the examination, a 100-point reticle with 50 lines was used coupled to the eyepiece of a conventional microscope [26]. The fraction area of normal alveoli (%) and alveolar collapse (%) were analyzed quantitatively by the point-counting technique, whereas the air-space enlargement was quantified by the mean linear intercept length of the distal air spaces. The bronchoconstriction index (*BCI*) was determined by counting the no. of points in the airway lumen and interceptions through the airway wall, using a reticulum and the following formula:

$$BCI = Airway_{lumen} \sqrt{Airway_{wall}} BCI = Airway_{lumen} \sqrt{Airway_{wall}}$$

## In silico assay

**Edition of the protein and FOR811A.** For the edition of the protein, a search for the three-dimensional model of the β subunit and the H-NOX domain (PDB: 4U99) was carried out in the Protein Database (PDB, https://www.rcsb.org). These structures were edited in PyMOL 2.0 (Schrödinger, LLC, Nova York, EUA) to add polar hydrogens and remove molecules of water. The FOR811A compound was designed in the MarvinSketch–v17.29 program (ChemAxon, Budapeste, Hungria). The analysis of the optimization of atomic geometry, energy minimization by the Density Functional Theory (DFT), and the refinement by calculating the charges of Gasteiger and Marsili [27] were performed in the AutoDockTools 1.5.6 program [28].

**Molecular docking.** To explore the drug-protein binding mechanisms, the Molegro Virtual Docker (MVD) was used. The docking parameters were those available in the MVD for anchoring (MolDock: anchoring score, rank score, and interaction energy scores) with information display (GRID). The fitting model had as its starting point the detection of the cavity in the region of the active site of H-NOX, taking into account the Heme group present in the structure. The grid settings (Search Space) obtained the following dimensions: X: -40.03; Y: -93.35; Z: 109.28 with a radius of 11 ångström (Å) to encompass both the distal and proximal site of the Heme group and the cysteine residue (Cys141). The MVD evolutionary algorithm was used for the benchmark, with the following data: number of executions, 10; population size, 50; maximum interaction, 2000; scale factor, 0.50; crossing rate, 90; and variation-based termination configuration. The program's performance was 500un with the return of 10 promising poses of the FOR811A compound. The simulation was performed in a rigid body system for H-NOX.

**Evaluation of interactions between FOR811A and protein.** The results were filtered based on the anchoring score, rank score, and interaction energy scores, with the poses with the lowest binding free energy (kcal/mol) [29]. The atoms and residues involved in the ligand's interactions with the receptor were observed by PyMOL 2.0 (Schrödinger, LLC, Nova York, USA) and Chimera.

## Statistical analysis

The data of pulmonary mechanics were evaluated in the One-Way test of variance (ANOVA), followed by the Bonferroni test, using the GraphPad® Prism software, version 7.0. The results were presented as mean ± standard error of the mean (SEM), and values of $p < 0.05$ were considered significant for the analysis.

## Results

### FOR811A decreased the pressure-volume curve (PV)

The pressure-volume curves (Fig 2A) for the untreated groups were higher in those with asthma (Ast+Sal) compared to healthy ones (Ctl+Sal), differing statistically ($p < 0.05$). Among the asthmatic animals, the PV was 1.68 mL in the animals treated with FOR811A (Ast+FOR) and 2.22 mL in the untreated group (Ast+Sal), showing that the metallocompound decreased the PV curve in 37.89% ($p < 0.05$) on asthma pathogenic condition. There was no statistical difference between healthy animals (Ctl+Sal) and those treated with FOR811A. These findings showed that this metallocompound changed the PV curve in asthmatic condition (Ast+FOR), making it similar to that observed in the control groups (Ctl+Sal; Ctl+FOR).

### FOR0811A maintained the static compliance (C$_{stat}$) normal

The static compliance (Fig 2B) was 0.073 mL/cmH$_2$O in the untreated asthmatic group (Ast +Sal), whereas in the untreated healthy group (Ctl+Sal) was 0.093 mL/cmH$_2$O, showing that asthma decreased the pulmonary C$_{stat}$ in 21.68% ($p < 0.05$). However, when treated with FOR811A, the C$_{stat}$ of the control (Ctl+FOR) and asthmatic (Ast+FOR) groups were similar ($p > 0.05$), even when compared to untreated healthy groups (Ctl+Sal). These findings showed

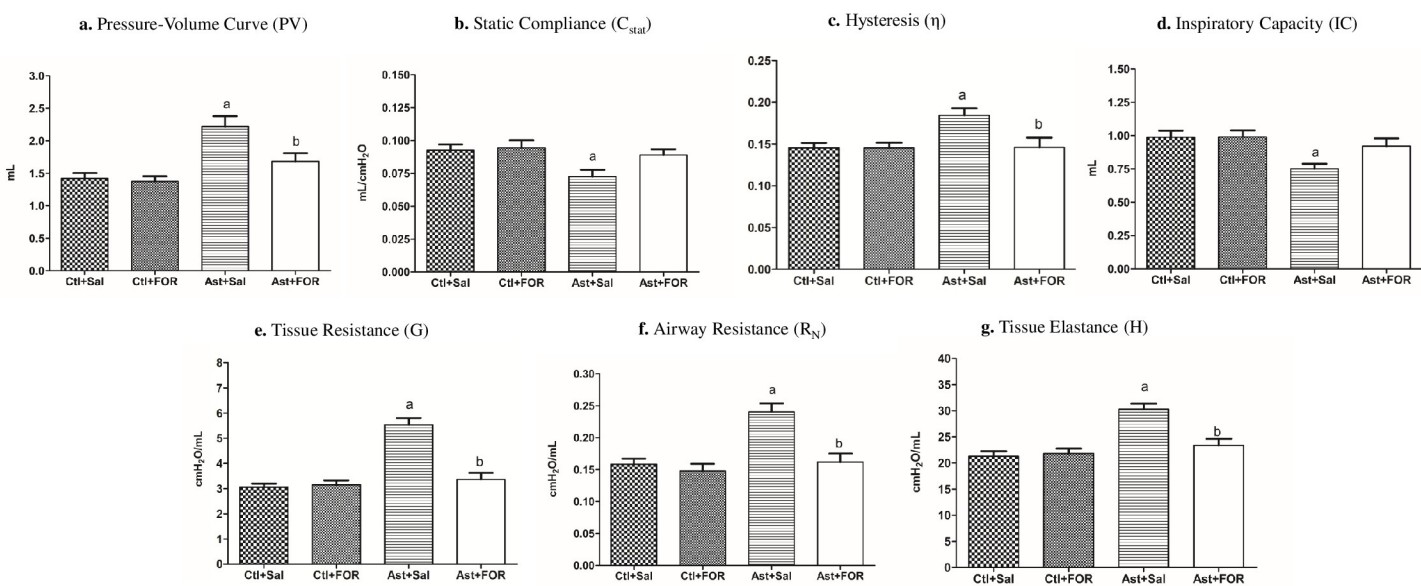

**Fig 2. Effect of the FOR811A, a ruthenium nitric oxide donor metallocompound, against experimentally induced asthma in a murine model.** The Swiss mice were divided into four groups ($n$ = 10, each): untreated (Ctl+Sal) and treated (Ctl+FOR) control groups, and untreated (Ast+Sal) and treated (Ast+FOR) asthmatic groups. Using an integrated platform for pulmonary mechanic assay, the pressure-volume curve area (PV; expressed in mL; **a**) and the behavior of static compliance (C$_{stat}$; expressed in mL/cmH$_2$O; **b**), hysteresis (η; **c**); inspiratory capacity (IC; expressed in mL; **d**); tissue resistance (G; expressed in cmH$_2$O.s/mL; **e**); airway resistance (R$_N$; expressed in cmH$_2$O.s/mL; **f**) and tissue elastance (H; expressed in cmH$_2$O.s/mL; **g**). Data were expressed as mean ± SEM. [a] $p < 0.05$ (Ast+Sal) vs. (Ctl+Sal); [b] $p < 0.05$ (Ast +FOR) vs. (Ast+Sal); [c] $p < 0.05$ (Ast+FOR) vs. (Ctl+Sal).

that the metallocompound improved the pulmonary $C_{stat}$ in treated asthmatic animals (Ast +FOR), remaining unchanged under normal conditions.

## FOR811A decreased the hysteresis (η)

Among untreated groups, the hysteresis (Fig 2C) was higher in animals with asthma (Ast+Sal) than in healthy animals (Ctl+Sal), showing that this pathogenic condition increased η in 42.46% ($p<0.05$). Among the asthmatic groups, the animals treated with FOR (Ast+FOR) presented 0.146 η, whereas in untreated animals (Asth+Sal) was 0.184 η. This finding showed that the metallocompound decreased the hysteresis by 32.59% during asthma ($p<0.05$). There was no statistical difference between the untreated control (Ctl+Sal) and the treated groups (Ctl +FOR; Ast+FOR).

## FOR811A maintained the Inspiratory Capacity (IC) normal

The inspiratory capacity (Fig 2D) of the untreated asthmatic group (Ast+Sal) was 0.751 mL, whereas in the untreated control group (Ctl+Sal) was 0.988 mL, showing that asthma decreased the IC in 24.04% ($p<0.05$). There was no statistically significant difference between the healthy treated group (Ctl+FOR) and the asthmatic treated group (Ast+FOR) when compared to the healthy untreated group (Ctl+Sal). These results showed that the metallocompound increased the IC in asthmatic condition (Ast+FOR), maintaining normal in a similar way to the control groups (Ctl+Sal; Ctl+FOR).

## FOR811A decreased the tissue resistance (G), airway resistance ($R_N$), and tissue elastance (H)

The tissue resistance (Fig 2E) of the untreated asthmatic group (Ast+Sal) was 5.54 cmH$_2$O.s/ mL, whereas in the healthy untreated group (Ctl+Sal) was 3.05 cmH$_2$O.s/mL, showing that asthma increased the G in 44.95% ($p<0.05$). However, the G in the treated asthmatic group (Ast+FOR) was 3.36 cmH$_2$O.s/mL, whereas in the untreated asthmatic group (Ast+Sal) was 5.54 cmH$_2$O.s/mL, showing that the metallocompound decreased the tissue resistance in 60.65% ($p<0.05$). There was no statistically significant difference between the treated healthy (Ctl+FOR) and the treated asthmatic (Ast+FOR) groups, and neither when both were compared to the untreated healthy group (Ctl+Sal). Based on these results, the metallocompound decreased tissue resistance, improving it even in asthmatic conditions.

The airway resistance (Fig 2F) in the untreated asthmatic group (Ast+Sal) was 0.241 cmH$_2$O.s/mL, whereas in the untreated healthy group (Ctl+Sal) it was 0.162 cmH$_2$O.s/mL, showing that asthma increased the $R_N$ in 67.2% ($p<0.05$). However, when comparing the $R_N$ of the treated asthmatic group (Ast+FOR), which was 0.162 cmH$_2$O.s/mL, it was observed that the metallocompound promoted a decrease in the airway resistance by 32.8% during asthma ($p<0.05$). There was no statistically significant difference between the healthy (Ctl+FOR) and asthmatic (Ast+FOR) groups that received the metallocompound, nor when both were compared to the untreated healthy group (Ctl+Sal). These data supported that the FOR811A significantly decreased the $R_N$ in an asthmatic condition, remaining under normal conditions.

The tissue elastance (Fig 2G) of the untreated asthmatic group (Ast+Sal) was 30.29 cmH$_2$O. s/mL, whereas in the untreated healthy group (Ctl+Sal) was 21.26 cmH$_2$O.s/mL, showing that the parameter H was increased by 29.81% ($p<0.05$) during asthma. However, the H in the treated asthmatic group (Ast+FOR) was 23.36 cmH$_2$O.s/mL, showing that the metallocompound decreased the tissue elastance in 22.82% during asthma condition ($p<0.05$). There was no statistically significant difference between the treated healthy (Ctl+FOR) and the treated asthmatic (Ast+FOR) groups, and neither when both were compared to the untreated healthy

**Table 1. Morphometric parameters in the respiratory mechanics of asthmatic and non-asthmatic Swiss mice after treatment with the nitric oxide metallo-donor FOR811A.**

| Experimental groups | Normal Alveoli (%) | Alveolar Collapse (%) | Mean Alveolar Diameter (μm) | Bronchoconstriction index (*BCI*) |
|---|---|---|---|---|
| Ctl+Sal | 90.03±2.36 | 9.97±2.36 | 44.56±3.31 | 2.05±0.21 |
| Ast+Sal | 71.58±6.22* | 28.42±6.22* | 35.54±5.56* | 1.99±0.17 |
| Ctl+FOR | 94.25±2.26 | 5.75±2.26 | 46.11±7.12 | 2.84±0.39 |
| Ast+FOR | 83.31±4.98 | 16.69±4.98 | 41.20±5.88 | 2.33±0.23 |

Values are mean ± SD of the following groups: control treated with saline solution (Ctl+Sal), asthmatic treated with saline solution (Ast+Sal), control treated with FOR811A (Ctl+FOR) and asthmatic treated with FOR811A (Ast+FOR). Data was collected in 10 matched fields per mouse. Values significantly different ($p < 0.05$) by one-way ANOVA followed by Student–Newman–Keuls test compared to the Ctl+Sal group (*), and no difference compared to Ctl+Sal group.

group (Ctl+Sal). In summary, the metallocompound decreased the pulmonary tissue elastance in asthmatic conditions, remaining under normal conditions.

## FOR811A decreased the alveolar collapse and kept the bronchoconstriction

According to Table 1, the control groups (Ctl+Sal and Ctl+FOR) showed a high percentage of normal alveolar pattern with few areas of alveolar collapse, presenting a similar mean alveolar diameter and a similar *BCI* of 2.05±0.21 (Ctl+Sal) and 2.84±0.39 (Ctl+FOR). For the untreated asthmatic group (Ast+Sal), a smaller normal alveolar area with a large area of the alveolar collapse was expected, resulting in a smaller mean alveolar diameter and a low rate of *BCI* of 1.99±0.17. Interestingly, the asthmatic group treated with the metallocompound (Ast+FOR) presented standards of normal alveoli, alveolar collapse, and mean alveolar diameter similar to those of the control groups, in which the *BCI* was 2.33±0.23. The reduction in the fractional area of collapsed alveoli (%), with less airway narrowing (BCI) in asthmatic animals by FOR811A (Ast+FOR) showed that this metallocompound attenuated the bronchoconstriction by promoting relaxation of smooth muscles. This action can be corroborated from data found in pulmonary mechanics, where the respiratory parameters of asthmatic animals treated with the metallocomposite were similar to those of the saline control group. This action likely occurred in the cysteine portion of the GCs enzyme, since these increase cGMP, which is important for smooth muscle relaxation.

## FOR811A interaction with the soluble guanylate cyclase enzyme

Through computational investigation of the interaction of the FOR811A on the geometric planes of the Heme group of the soluble *guanylate cyclase* (sGC), it was observed that the distal portion of this group was strongly associated to the binding of the exogenous molecule (Fig 3A) in comparison to the proximal portion. Analysis of the drug-protein binding conformation, types of interactions, residues involved in the stabilization of the system stabilization, mechanisms, and conformational changes of the target enzyme in the FOR811A–H-NOX were performed. The results showed that the MolDock score was 119.282, the rank score was 1993.62 and the interaction score was –1.0376 during the inactive state of H-NOX. At the radius of 6 Å, the presence of six amino-acids–Ile5, Phe69, Gly70, Leu73, Leu77, Leu145 –was defined, which interacted sterically with the Cys141 residue, performing hydrogen bonding interaction with atoms ID N:17, N49 at 1.34 Å (Fig 3B and 3C).

## Discussion

This study used mice to assess bronchial asthma, as an experimental model that allows a wide allergic response similar to human asthma, including acute and late-phase responses. By using

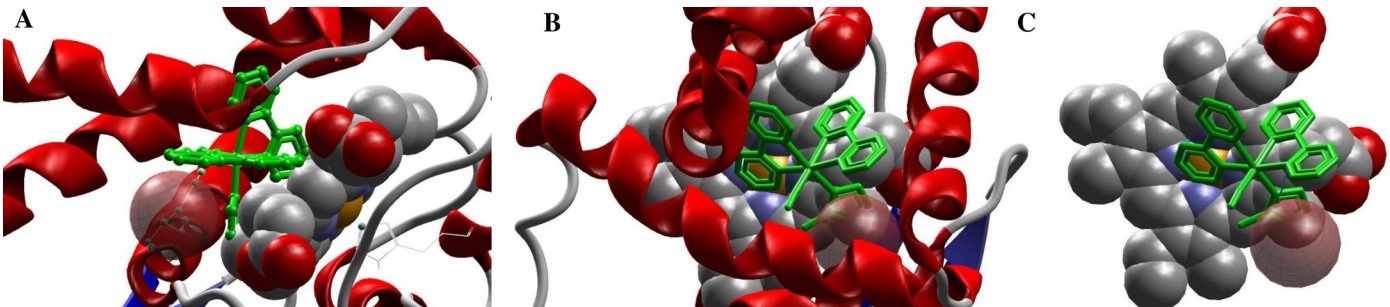

**Fig 3. Interaction of the FOR811A compound, a ruthenium nitric oxide donor metallocompound with anti-asthmatic potential, on the Heme portion of the soluble *guanylate cyclase* (sGC) enzyme.** Through the exploration of the potential drug-protein binding mechanisms by the molecular docking method, it was observed that FOR811A bound strongly to the distal portion of the Heme group of sGC, specifically interacting with the residue Cys141 (**a**). In this study with H-NOX, FOR811A was shown to interact with the thiol portion of the cysteine residue at position 141 (Cys 141) (**b**, **c**).

the methods proposed in the literature [30, 31], ovalbumin was used as an antigenic protein to promote systemic sensitization in experimental animals, followed by the inhalation challenge. Thus, for the first time, it was possible to carry out tests regarding the anti-asthmatic effect of the FOR811A on the pulmonary mechanics of Swiss mice successfully.

The analysis of the pressure-volume curve was highly evident in the untreated asthmatic group (Ast+Sal), due to changes in the distribution of the surfactant on the alveolar surface and the presence of alveolar edema in these animals. In turn, this parameter was decreased in those asthmatic animals treated with the metallocompound (Ast+FOR), as well as in the untreated (Ctl+Sal) and treated (Ctl+FOR) control groups. It is known that asthma causes hyperinflation that alters the respiratory mechanics, increasing the functional residual capacity and rectifying the diaphragm position, due to the reduction in the strength of the respiratory muscles [32]. Because of lower muscle strength, due to hyperinflation and greater respiratory effort, the animals are at risk of muscle fatigue and consequent dyspnea and respiratory failure [33]. Therefore, asthma leads to a significant loss of respiratory muscle strength and mass [34], as seen in untreated asthmatic animals (Ast+Sal).

To assess the degree of lung extension for each increase in transpulmonary pressure ($C_{stat}$) and the maximum volume of inspiration after a baseline expiration (IC), the equation proposed by Salazar and Knowles was calculated [35] during the upper half of the expiatory branch of the PV curve to adjust these parameters. The results indicated that the untreated asthmatic animals (Ast+Sal) presented low values of $C_{stat}$ and IC, in comparison to the remaining groups, as well as high values of tissue elastance (H). These data corroborate with pulmonary physiology, in which the static compliance is inversely proportional to tissue elastance [36], showing that this group had high difficulty in breathing inspiration, in addition to stiffening of lung tissue. Interestingly, these parameters–$C_{stat}$, CI e H–were statistically identical for the remaining groups (Ast+FOR; Ctl+FOR; Ctl+Sal), showing that the metallocompound improved the capacity for inspiration even in asthmatic conditions, as similarly to healthy animals.

The friction resulting from the detachment of the pulmonary, thoracic, and muscular tissues leads to tissue resistance (G) during the inspiration and expiration process [36], which was highly observed in untreated asthmatic animals (Ast+Sal). This finding results from hyper-responsiveness during allergic inflamed airways, as in asthma. This process occurs because there is a thickening of the airway mucosa and a greater propensity for them to close, even without increasing the degree of shortening of the airway smooth muscle [37].

In the case of airway resistance, which measures the degree of difficulty that airflow must move during the trachea-bronchial path [36], it was also possible to observe high values in the untreated asthmatic group (Ast+Sal) compared to the remaining groups ($p < 0.05$). Considering that this parameter is inversely proportional to airflow [36], this finding showed that the untreated asthmatic group (Ast+Sal) had less airflow, resulting in greater difficulty for the airflow to move along the lung tree.

This occurred because, during the pathogenesis of asthma, the high contractility of smooth muscle in the airways contributes to the increase in airway resistance [38]. Due to the reduction in the elastic recoil of the lung and the rupture of fibers, the destruction of alveolar attachments occurs, which limits airflow, causing an increase in airway resistance. Therefore, the air trapping and the new pulmonary morphology reduce the surface available for gas exchange [39]. Interestingly, the treated asthmatic group (Ast+FOR) showed low values of $R_N$, as well as the control groups (Ctl+Sal; Ctl+FOR), revealing the maintenance of pulmonary architecture and smooth muscle relaxation tracheobronchial by FOR811A. This result showed that the metallocompound functioned similarly to conventional treatments, such as bronchodilators and corticosteroids [7, 8], which aim at reducing the airway resistance [6], reversing the asthma crisis.

Tissue resistance (G) and tissue elastance (H) are related to the intrinsic properties of the tissue, and the analysis of these parameters is not as simple as airway resistance ($R_N$). This may be due to the alteration of the rheological properties of the tissue [40] or the influence of the narrowing of the airways on such parameters–G and H. Thus, it would result in a distortion of the lung parenchyma with the closure of small airways, constituting an effectively smaller lung, however, with a proportionally greater tissue elastance [37]. This directly proportional relationship was observed, since the untreated asthmatic animals (Ast+Sal) presented high values of G and H, as a result of the high $R_N$, unlike the other groups (Ast+FOR; Ctl+Sal; Ctl+FOR). Therefore, the low values of G, H, and $R_N$ revealed that FOR811A maintained the pulmonary parenchyma and airway opening, even after the experimental induction of allergic asthma.

Hysteresis ($\eta$) occurs due to the resistance of the lung tissue [40], and it is calculated based on the relationship of parameters G and H. Therefore, hysteresis is determined by the elastic forces of the respiratory musculature and by the elastic forces caused by the surface tension of the liquid that lines the inner walls of the alveoli and other air spaces of the lung [36]. Hysteresis is also indicative of ventilatory heterogeneity since the value of $\eta$ increases proportionally as the lung becomes mechanically heterogeneous [36]. Thus, the pathogenic asthma conditions, as in the untreated asthmatic group (Ast+Sal), cause a greater difference between the pulmonary inflation and deflation curve [40], due to the high resistance of the lung tissue. On the other hand, the treated asthmatic animals (Ast+FOR) showed normal values of $\eta$, similarly to the control groups (Ctl+Sal; Ctl+FOR), showing that FOR811A satisfactorily reduced pulmonary heterogeneity and atelectasis in the experimental animals.

The ruthenium-based compound normalized the pulmonary conditions during asthma, since the values of $C_{stat}$, IC, $R_N$, G, H, and $\eta$ were similar to the control groups. Additionally, the morphometric data supported that FOR811A decreased the area of alveolar collapse, allowing the bronchoconstriction, similarly to the control groups. In this way, our findings supported that FOR811A is indeed promising for the development of an anti-asthmatic pharmaceutical product. This anti-asthmatic effect could be explained by the activation of the enzyme sGC through its cysteine residue within the heme site, causing also the release of NO by the metallocompound, and further studies should be performed to confirm this hypothesis. It is known that NO can control the proliferation and differentiation of muscle cells through the activation of sGC and, consequently, increased levels of cGMP. On the other hand, it is also believed that NO can control some of these events through an independent mechanism,

such as the formation of S-nitrosothiols, which can modulate the formation of muscle fibers [41].

In this way, the anti-asthmatic effect of FOR811A may be due to the donation of NO from this metallodrug, activating sGC through the rupture of histidine in the proximal site that is linked to iron, a well described process [42]. Beyond that and according to the results of molecular docking, FOR811A interacts with the enzyme sGC in the distal portion of the heme group, more specifically with the cysteine residue Cys141. It is possible that this interaction favors an *in situ* nitrosylation of this residue, where the $Ru^{II}$-$NO^{+}$ moiety of FOR811A is prompt for this reaction. Indeed, full activation of sGC was shown to require not only heme Fe-NO formation but also nitrosylation of cysteine for full sGC activation [43]. Thus, NO from this compound along with an *in situ* nitrosilation of cysteine can fully activate sGC and promote this event [44]. That being said, it is believed that the anti-asthmatic activity of FOR811A was due to its action on the soluble *guanylate cyclase* enzyme, activating it to increase the production of cGMP. In turn, this would have acted directly on the smooth muscle cells of the airways, causing their relaxation, improving the mechanical pulmonary conditions in animals experimentally induced to asthma and treated with FOR811A (Ast+FOR).

## Conclusions

The use of the nitrosyl-ruthenium *cis*-$[Ru(bpy)_2(2\text{-MIM})(NO)](PF_6)_3$, called FOR811A, allowed the relaxation of bronchial smooth muscles, improving the respiratory mechanics caused by the induction of asthma. Given the smooth muscle relaxation, this potential conferred a protective effect of asthmatic conditions, possibly due to its dual action on the cysteine and heme portion of the sGC enzyme.

## Supporting information

**S1 Fig. Scheme of the integrated platform for data collection regarding pulmonary mechanics measurements.** Through a carbogen cylinder **(A)**, oxygenation was maintained at a ratio of 95%:5% (O2:CO2). An air purification unit **(B)** was attached to the equipment responsible for maintaining the heating and humidification of the air **(C)** and the depressurizer **(D)**, mechanical fan for small animals **(E)** and ultrasonic nebulizer **(F)**. Also, a reservoir containing bronchoconstrictor **(G)** and a bed with heating support **(H)** for maintaining the body at a temperature of 37˚C were available.
(TIF)

## Author Contributions

**Conceptualization:** Fladimir de Lima Gondim.

**Data curation:** Paula Priscila Correia Costa.

**Formal analysis:** Stefanie Bressan Waller, Helena Serra Azul Monteiro.

**Investigation:** Gilvan Ribeiro dos Santos, Fladimir de Lima Gondim, Daniel Silveira Serra, Francisco Sales Ávila Cavalcante, Florêncio Sousa Gouveia Júnior, Valdir Ferreira de Paula Júnior, Eduardo Henrique Silva Sousa, Luiz Gonzaga de França Lopes, Wesley Lyeverton Correia Ribeiro, Helena Serra Azul Monteiro.

**Methodology:** Paula Priscila Correia Costa, Gilvan Ribeiro dos Santos, Fladimir de Lima Gondim, Daniel Silveira Serra, Francisco Sales Ávila Cavalcante, Florêncio Sousa Gouveia Júnior, Valdir Ferreira de Paula Júnior, Eduardo Henrique Silva Sousa, Luiz Gonzaga de França Lopes, Wesley Lyeverton Correia Ribeiro, Helena Serra Azul Monteiro.

**Resources:** Florêncio Sousa Gouveia Júnior, Eduardo Henrique Silva Sousa, Luiz Gonzaga de França Lopes.

**Supervision:** Valdir Ferreira de Paula Júnior, Wesley Lyeverton Correia Ribeiro.

**Validation:** Paula Priscila Correia Costa, Gilvan Ribeiro dos Santos, Daniel Silveira Serra, Francisco Sales Ávila Cavalcante, Florêncio Sousa Gouveia Júnior, Valdir Ferreira de Paula Júnior, Luiz Gonzaga de França Lopes.

**Visualization:** Eduardo Henrique Silva Sousa, Wesley Lyeverton Correia Ribeiro, Helena Serra Azul Monteiro.

**Writing – original draft:** Paula Priscila Correia Costa, Stefanie Bressan Waller.

**Writing – review & editing:** Stefanie Bressan Waller, Helena Serra Azul Monteiro.

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
