## [Decision Letter · Decision Letter 0]

3 Nov 2020

PONE-D-20-30462

Anti-asthmatic effect of nitric oxide metallo-donor FOR811A [cis-[Ru(bpy)2(2-MIM)(NO)]3+(PF6)3] in the respiratory mechanics of Swiss mice

PLOS ONE

Dear Dr. Stefanie Waller,

Thank you for submitting your manuscript to PLOS ONE. After careful consideration, we feel that it has merit but does not fully meet PLOS ONE’s publication criteria as it currently stands. Therefore, we invite you to submit a revised version of the manuscript that addresses the points raised during the review process.

This paper is interesting but it needs a significant revision following reviewers' suggestions.

In conclusion the author should not use the term respiratory failure which means alteration in gas exchange but it is correct the term respiratory mechanics.

In addition, the authors should cite at least 1 or 2 reviews on the role of nitric oxide in respiratory system, especially in a model of bronchoconstriction.

We look forward to receiving your revised manuscript.

Kind regards,

Fabio Luigi Massimo Ricciardolo

Academic Editor

PLOS ONE

Journal Requirements:

2. At this time, we request that you  please report additional details in your Methods section regarding animal care, as per our editorial guidelines:

(1) Please state the original source of mice used in the study

(2) Please describe any steps taken to minimize animal suffering and distress from the asthma challenge, such as by administering anaesthesia

(3) Please describe the post-challenge care received by the animals, including the frequency of monitoring and the criteria used to assess animal health and well-being.

Thank you for your attention to these requests.

3. In your Methods section, please state the source of the ovalbumin used in your study.

4. Please ensure your Methods and reagents are be described in sufficient detail for another researcher to reproduce the experiments described. Specifically, please provide further details on the synthesis of FOR811A, including a short summary of the synthesis conditions, and any chemical characterisation that was performed.

5. To comply with PLOS ONE submissions requirements, please provide the method of euthanasia in the Methods section of your manuscript.

Additional Editor Comments (if provided):

This paper is interesting but it needs a significant revision following reviewers' suggestions.

In conclusion the author should not use the term respiratory failure which means alteration in gas exchange but it is correct the term respiratory mechanics.

In addition, the authors should cite at least 1 or 2 reviews on the role of nitric oxide in respiratory system, especially in a model of bronchoconstriction.

Reviewers' comments:

Reviewer's Responses to Questions

**Comments to the Author**

1. Is the manuscript technically sound, and do the data support the conclusions?

Reviewer #1: Partly

Reviewer #2: Partly

2. Has the statistical analysis been performed appropriately and rigorously? 

Reviewer #1: No

Reviewer #2: Yes

3. Have the authors made all data underlying the findings in their manuscript fully available?

Reviewer #1: Yes

Reviewer #2: Yes

4. Is the manuscript presented in an intelligible fashion and written in standard English?

Reviewer #1: Yes

Reviewer #2: No

5. Review Comments to the Author

Reviewer #1: In their manuscript, Costa and collaborators demonstrated that the administration of FOR811A, a compound containing ruthenium, is able to preserve inspiratory abilities in a mouse model of asthma.

However, the authors mention in their introduction that another ruthenium-containing compound (TERPY) is already capable of controlling asthma. So what are the advantages and disadvantages of FOR811A compared to TERPY?

Why did the authors choose to use female Swiss mice? Have any sampling power studies been performed? Furthermore, the ways of sacrificing animals are not described in the manuscript.

Finally, the interaction studies between molecules seem to have been done exclusively through software simulation. Would it be possible to carry out an experimental analysis in order to actually verify the identified interactions and possibly exclude others that were not detected?

Reviewer #2: Costa and coworkers assessed the effect of a new metallodrug known as FOR811A on allergic asthma murine model. The authors revealed improvements in the pulmonary mechanics in mice treated with FOR811A suggesting that the effectiveness of FOR811A is due to the interaction between the drug and the soluble guanylate cyclase (sGC). Thus resulting in a significant increase in the production of cyclic guanosine monophosphate (cGMP).

It is an interesting study, but the following should be addressed:

Major comments:

1) In my opinion, the abstract need to be rewritten because of its confusing structure.

2) The authors performed their experiments on 40 Swiss female mice. What was the reasoning behind the choice to use only female mice as a model?

3) In the results section (page 11, lines 301-302) the authors state: “These findings highlighted that the metallodrug FOR811A allowed the bronchoconstriction activity because it decreased alveolar collapse during asthma.” Please explain the meaning of the sentence more clearly.

4) In the discussion section Costa et al., declare that the anti-asthmatic effect of FOR811A can be explained by the activation of the enzyme soluble guanylate cyclase (sGC) in the cysteine portion, causing the release of NO by the metallodrug (page 14, lines 421-423). The data reported in this article do not support the conclusion provided by the authors. They should attempt to validate this hypothesis with experimental data. The authors could measure the sGC activity by enzyme assay.

5) The paper needs to be proofread by a native English speaker to correct several inaccuracies in English.

Minor comments:

1) In the “Experimental Design and Treatments” the authors should define the meaning of “Sal” (page 4, line 107).

2) Please rewrite Figure Legend 1 because is confused and unclear.

3) In the caption of Table 1 the authors state: “Values significantly different (p<0.05) by one-way ANOVA followed by Student–Newman–Keuls test compared to the Ctl+Sal group (*), and no difference compared to Ctl+Sal group (a).” In my opinion, the authors could avoid indicating the absence of statistically significant differences and remove “a”.

4) Figure 3 (legend): the authors should describe the figure more clearly.

6. PLOS authors have the option to publish the peer review history of their article (what does this mean?). If published, this will include your full peer review and any attached files.

Reviewer #1: No

Reviewer #2: No

---

## [Author Response · Author response to Decision Letter 0]

22 Jan 2021

Dear,

The authors are grateful for the suggestions. Please, the responses below.

Editor comments: 

Answer: The authors are grateful for the suggestions. Furthermore, all changes suggested by the Editor are presented in PINK in the manuscript. Additionally, a structural change was made to the subtitles of the results, where “FOR811A decreased the Tissue Resistance (G)” (line 257), “FOR811A decreased the Airway Resistance (RN)” (line 268), and “FOR811A decreased the Tissue Elastance (H)” (line 279) were unified to “FOR811A decreased the Tissue Resistance (G), Airway Resistance (RN), and Tissue Elastance (H)” (lines 283-284).

2. At this time, we request that you please report additional details in your Methods section regarding animal care, as per our editorial guidelines:

(1) Please state the original source of mice used in the study

Answer: The animals were obtained from the central vivarium of Federal University of Ceará (UFC, Fortaleza, Ceará, Brazil). This information is shown in the text in: “The animals were obtained from the Central Bioterium (Federal University of Ceará, Fortaleza/CE, Brazil)” (lines 110-111).

(2) Please describe any steps taken to minimize animal suffering and distress from the asthma challenge, such as by administering anaesthesia

Answer: In order to minimize animal suffering and distress from the asthma challenge, all animals received Tramadol as an analgesic opioid. Therefore, the following sentence has been added to the text: “All animals also received tramadol (5 mg/kg/8h, Cronidor 2%®, Agener União Saúde Animal Ltda., São Paulo/SP, Brazil/SP, Brazil) as an analgesic method to minimize pain, suffering and distress” (lines 134-136).

(3) Please describe the post-challenge care received by the animals, including the frequency of monitoring and the criteria used to assess animal health and well-being.

Answer: Right. The requested information has been added in the text in the following sentence: “Additionally, the post-challenge care included the monitoring of the animals twice a day by veterinarians (8:00 am and 5:00 pm) for possible behavioral changes through the application of the Grimmace Scale. The researchers were previously trained to apply the "humanitarian endpoint", according to the Brazilian legislation, if any animal had its welfare compromised. Normal interaction with other animals, amount of feeding, and volume of water ingested were also monitored” (lines 136-141).

3. In your Methods section, please state the source of the ovalbumin used in your study.

Answer: Right. The sentence “ovalbumin (100 μg, dissolved in 5 mg of aluminum hydroxide – AlOH)” (lines 110-111) was updated to “ovalbumin (Sigma-Aldrich®, St. Louis/MO, USA; 100 μg, dissolved in 5 mg of aluminum hydroxide – AlOH)” (lines 126-127).

4. Please ensure your Methods and reagents are described in sufficient detail for another researcher to reproduce the experiments described. Specifically, please provide further details on the synthesis of FOR811A, including a short summary of the synthesis conditions, and any chemical characterisation that was performed.

Answer: Right. This information has been added in the following sentence: “Briefly, 0.4 mmol of the precursor cis-[Ru(bpy)2Cl2] was reacted with equimolar amount of 2-methylimidazole in ethanol under reflux and magnetic stirring. After 2 hours, an equimolar amount of NaNO2 dissolved in water was added, keeping the system in the previous conditions for 2 more hours. After that, the solvent was removed through rotary evaporation and the crude product was mixed with a 10% HPF6 solution, giving the desired product as a light orange solid. Yield: 45%. HRESI-MS (+): [M – 3PF6]2+ theoretical: 263,0459 (C24H22N7ORu2+); experimental: 263,0459. Elemental analysis: Theoretical (C24H22F18N7OP3Ru): C, 30,01; H, 2,31; N, 10,21%. Experimental: C, 30,06; H, 2,38; N, 10,35%.” (lines 95-103).

5. To comply with PLOS ONE submissions requirements, please provide the method of euthanasia in the Methods section of your manuscript.

Answer: The method of euthanasia was provided in the text. in order to ease the reading, the sentence “Finally, on day 30, all animals were subjected to sedative and anesthetic induction with ketamine (10 mg/kg) associated with xylazine (2 mg/kg), both intraperitoneally, to allow the assessment of pulmonary mechanics measurements.” (lines 122-124) was updated to “Finally, on day 30, the association between 10% ketamine hydrochloride (300 mg/kg, Cetamin®, Syntec, São Paulo/SP, Brazil) and α2-adrenergic receptor agonists 2% xylazine hydrochloride (30 mg/kg - Sedanew®, Vetnil, São Paulo/SP, Brazil) was used for the euthanasia of mice by anesthetic overdose.” (lines 145-148).

Answer: You are right. The ethics statement presented at the end of the manuscript has been removed. Currently, said information is only shown in the Methods section.

Additional Editor Comments (if provided):

This paper is interesting but it needs a significant revision following reviewers' suggestions.

In conclusion the author should not use the term respiratory failure which means alteration in gas exchange but it is correct the term respiratory mechanics.

Answer: …You are right. The text was verified, and the term “respiratory mechanics” was preserved. As such, the sentence “…and improved respiratory failure during asthma …” (line 44) has been updated to “…and improved respiratory mechanics during asthma …” (line 41), as well as the sentence “…improving the respiratory failure caused by the …” (lines 443-444) has been updated to “…improving the respiratory mechanics caused by the …” (lines 479-480).

In addition, the authors should cite at least 1 or 2 reviews on the role of nitric oxide in respiratory system, especially in a model of bronchoconstriction.

Answer: Right. The role of nitric oxide in bronchoconstriction condition was briefly described, as shown by Karamaoun et al., 2016 (Modeling of the Nitric Oxide Transport in the Human Lungs. Front Physiol. 7:255. doi: 10.3389/fphys.2016.00255). Hence, the sentence “…against excessive bronchoconstriction [11,12]. However, NO of endogenous origin has …” (lines 71-72) has been updated to “…against excessive bronchoconstriction [11,12]. Having said that, the conditions of bronchoconstriction are modelled by NO, that acts as bronchodilator in the human lungs [13]. However, NO of endogenous origin has …” (lines 68-70). Additionally, all references have been updated in terms of numbering.

Reviewer #1: 

In their manuscript, Costa and collaborators demonstrated that the administration of FOR811A, a

compound containing ruthenium, is able to preserve inspiratory abilities in a mouse model of asthma.

However, the authors mention in their introduction that another ruthenium-containing compound (TERPY) is already capable of controlling asthma. So what are the advantages and disadvantages of FOR811A compared to TERPY?

Answer: Thank you for the suggestion. All changes have been presented in YELLOW in the manuscript. In a study evaluating the participation of endogenous nitric oxide (NO) under the relaxation caused by FOR811A under aortic rings of Wistar rats, TERPY presented negative modulation with loss of potency under intact endothelium and preparations devoid of endothelium, unlike FOR811A, which maintained power and effectiveness (BONAVENTURA et al., 2009). This difference was justified by the fact that, in TERPY, the metabolite [Ru(H2O)(bdq)(terpy)]2+ is released along with NO, inducing oxidation of the BH4 cofactor, forming dihydrobiopterine (BH2) and biopterin, which may be responsible for uncoupling nitric oxide endothelial synthase (eNOS). In the vascular environment, eNOs can have a profound effect on the bioavailability of NO, decreasing its production and increasing the production of superoxides (BONAVENTURA et al., 2009; POTJE et al., 2014). In another study with ruthenium complex [Ru(terpy)(bdq)NO+]3+ (TERPY), Bonaventura et al. (2009) observed negative modulation in preparations that contained indomethacin, indicating the involvement of prostanoids in the vasodilator mechanism of this compound. The increase in cGMP levels and the simultaneous activation of PKG reduce the intracellular Ca2+ concentration by different mechanisms, such as the activation of K+ channels and the inhibition of Ca2+ L-type channels (MERY et al., 1991) or direct activation of these channels for K+ by NO (BOLOTINA et al., 1994) indirectly by cGMP (ROBERTSON et al., 1993). These findings demonstrated, therefore, that indomethacin does not modify the potency and effectiveness of FOR811A in relaxing the aortic rings, unlike TERPY, constituting an advantage over the other ruthenium complex. In addition, the opening of these channels in vascular smooth muscle cells causes the efflux of this ion, which generates membrane hyperpolarization. This reduces the influx of Ca2+ via voltage operated channels, with consequent vasodilation (JACKSON, 2017). Thus, the contribution of channels to K+ in relaxation induced by FOR811A was studied using a non-selective blocker of these channels, tetraethylammonium (10 mmol / L). This blocks different types of K+ channels with different degrees of effectiveness (PEREIRA et al., 2013). In preparations pre-incubated with this blocker, there was a shift in the concentration-effect curve to the left, demonstrating a concentration-dependent increase in the power of FOR811A. In TERPY, the concentration-effect curve is shifted to the right, showing a concentration-dependent reduction in TERPY power (Bonaventura 2007). However, the results of tetraethylammonium in FOR811A preparations are a type of finding not yet described in ruthenium complexes.

● BOLOTINA, V. M., et al. Nitric oxide directly activates calcium-dependent potassium channels in vascular smooth muscle cells. Nature, v. 368, p. 850-853, 1994.

● BONAVENTURA, D. et al. Endothelium negatively modulates the vascular relaxation induced by nitric oxide donor, due to uncoupling NO synthase. J. Inorg. Biochem., v. 103, n. 10, p. 1366-1374, 2009.

● JACKSON, W. F. Potassium channels in regulation of vascular smooth muscle contraction and

● growth. Adv. Pharmacol., v. 78, p. 89-144, 2017.

● POTJE, S. R. et al. Mechanisms underlying the hypotensive and vasodilator effects of [Ru(terpy)(bdq)NO]3+, a nitric oxide donor, differ between normotensive and spontaneously hypertensive rats. Eur. J. Pharmacol., v. 741, p. 222-229, 2014.

● MERY, P. F.; LOHMANN, S. M.; FISCHMEISTER, R. Ca2+ current is regulated by cyclic GMP dependent protein kinase in mammalian cardiac myocytes. Proc. Natl. Acad. Sci. U.S.A., v. 88, n. 4, p. 1197-1201, 1991.

● ROBERTSON, B. E. et al. cGMP-dependent protein kinase activates Ca-activated K channels in cerebral artery smooth muscle cells. Am. J. Physiol., v. 265, n. 1 (pt. 1), c. 299-303, 1993.

Why did the authors choose to use female Swiss mice? Have any sampling power studies been performed?

Answer: The use of Outbred mice (Swiss mice) was used due to the fact that the human species is not consanguineous, in order to facilitate the extrapolation of results in humans (Olson and Graham, 2014 - Animal Models in Pharmacogenomics, Chapter 5, Editor(s): Sandosh Padmanabhan, Handbook of Pharmacogenomics and Stratified Medicine, Academic Press, 2014, Pages 73-87). The use of Diversity Outbred (DO) mice is a new population, whose level of genetic diversity is at the same level as humans and non-human primates (Svenson et al., 2012 - High-resolution genetic mapping using the Mouse Diversity Outbred population. Genetics 2014;190:437–447. doi:10.1534/genetics.111.132597). In relation to sex, it has been shown that female mice are more susceptible to severe allergic inflammation than males (Blacquière et al. 2010 - Airway Inflammation and Remodeling in Two Mouse Models of Asthma: Comparison of Males and Females. Int Arch Allergy Immunol 2010;153:173-181. doi: 10.1159/000312635). This may be related to the low levels of TGF- β1 observed in female mice (Letterio and Roberts, 1988 - Regulation of immune responses by TGF. Annu Rev Immunol;16:137–161). In order to reduce the number of animals used in our study, we opted for an experimental model in which the inflammation was likely more exacerbated for us to have more reliable results on the effect of FOR811A in experimental asthma. The results obtained with the use of Swiss mice allowed us to understand the anti-asthmatic effect of FOR811A, and set the stage for efforts to discover this compound, which are underway.

 For text improvements, the sentence “...and commercial diet and water ad libitum. All procedures...” (lines 102-103) has been updated to “…and commercial diet and water ad libitum. This murine model was chosen because it does not show consanguinity, similarly to humans [19, 20], and because female mice seem to be more sensitive to develop allergic inflammation [21, 22]. All procedures …” (lines 113-116).

Furthermore, the ways of sacrificing animals are not described in the manuscript.

Answer: The method of euthanasia was provided in the text in PINK. The sentence “Finally, on day 30, all animals were subjected to sedative and anesthetic induction with ketamine (10 mg/kg) associated with xylazine (2 mg/kg), both intraperitoneally, to allow the assessment of pulmonary mechanics measurements.” (lines 122-124) has been updated to “Finally, on day 30, the association between 10% ketamine hydrochloride (300 mg/kg, Cetamin®, Syntec, São Paulo/SP, Brazil) and α2-adrenergic receptor agonists 2% xylazine hydrochloride (30 mg/kg - Sedanew®, Vetnil, São Paulo/SP, Brazil) was used for the euthanasia of mice by anesthetic overdose.” (lines 145-148).

Finally, the interaction studies between molecules seem to have been done exclusively through software simulation. Would it be possible to carry out an experimental analysis in order to actually verify the identified interactions and possibly exclude others that were not detected?.

Answer: Unfortunately, due to operational limitations, it is not possible to perform such tests. However, the increase in cGMP expected through the interactions seen in docking was confirmed in experiments carried out by our group: Silveira (2019) and Alves (2018).

● ALVES, N.T.Q. Renal effects of ruthenium complexes and their action in the protection of acute injury induced by ischemia and reperfusion. Thesis, Post-Graduation Program in Pharmacology, Federal University of Ceará, 2018. 109 p. [in Portuguese]

● SILVEIRA, J.A.M. Pharmacological characterization of the vasodilating activity of new ruthenium complexes containing imidazole derivatives. Thesis, Post-Graduation Program in Pharmacology, Federal University of Ceará, 2019. 147 p. [in Portuguese]

Reviewer #2: 

Costa and coworkers assessed the effect of a new metallodrug known as FOR811A on allergic asthma murine model. The authors revealed improvements in the pulmonary mechanics in mice treated with FOR811A suggesting that the effectiveness of FOR811A is due to the interaction between the drug and the soluble guanylate cyclase (sGC). Thus resulting in a significant increase in the production of cyclic guanosine monophosphate (cGMP). It is an interesting study, but the following should be addressed:

Answer: Thanks for the suggestions. All changes were performed in GREEN in the manuscript. 

Major comments:

1) In my opinion, the abstract needs to be rewritten because of its confusing structure. 

Answer: Right. We updated the abstract for improvements. Please, see abstract with correction in green color (lines 22-42).

2) The authors performed their experiments on 40 Swiss female mice. What was the reasoning behind the choice to use only female mice as a model?

Answer: The use of female mice was based to the greater susceptibility to develop severe allergic inflammation than males (Blacquière et al. 2010 - Airway Inflammation and Remodeling in Two Mouse Models of Asthma: Comparison of Males and Females. Int Arch Allergy Immunol 2010;153:173-181. doi: 10.1159/000312635). This may be related to low levels of TGF- β1 observed in female mice (Letterio and Roberts, 1988 - Regulation of immune responses by TGF- $ . Annu Rev Immunol;16:137–161). For text improvements, the sentence “...and commercial diet and water ad libitum. All procedures...” (lines 102-103) has been updated to “…and commercial diet and water ad libitum. This murine model was chosen because it does not show consanguinity, similarly to humans [19, 20], and because female mice seem to be more sensitive to develop allergic inflammation [21, 22]. All procedures …” (lines 113-116). This change was highlighted in YELLOW in the text.

3) In the results section (page 11, lines 301-302) the authors state: “These findings highlighted that the metallodrug FOR811A allowed the bronchoconstriction activity because it decreased alveolar collapse during asthma.” Please explain the meaning of the sentence more clearly.

Answer: Right. The finding indicated that the FOR811A attenuated the bronchoconstriction activity, and this can be explained by the lower narrowing of the airways (Table 1) probably because it prevented the inflammatory process. The morphometric study of the pulmonary parenchyma showed a reduction in the fractional area of collapsed alveoli (%), with less airway narrowing (BCI), showing that FOR811A attenuated the bronchoconstriction of asthmatic animals, by promoting relaxation of smooth muscles. This action probably occurred in the cysteine portion of the GCs enzyme, since this increases cGMP, which is important for smooth muscle relaxation. For text improvements, the sentence “These findings highlighted that the metallodrug FOR811A allowed the bronchoconstriction activity because it decreased alveolar collapse during asthma.” (lines 301-302) has been updated to “The reduction in the fractional area of collapsed alveoli (%), with less airway narrowing (BCI) in asthmatic animals by FOR811A (Ast+FOR) showed that this metallocompound attenuated the bronchoconstriction by promoting relaxation of smooth muscles. This action can be corroborated from data found in pulmonary mechanics, where the respiratory parameters of asthmatic animals treated with the metallocomposite were similar to those of the saline control group. This action probably occurred in the cysteine portion of the GCs enzyme, since these increase cGMP, which is important for smooth muscle relaxation” (lines 326-333).

4) In the discussion section Costa et al., declare that the anti-asthmatic effect of FOR811A can be explained by the activation of the enzyme soluble guanylate cyclase (sGC) in the cysteine portion, causing the release of NO by the metallodrug (page 14, lines 421-423). The data reported in this article do not support the conclusion provided by the authors. They should attempt to validate this hypothesis with experimental data. The authors could measure the sGC activity by enzyme assay.

Answer: You are right. Unfortunately, due to operational limitations, it was not possible to measure cGMP during the period of this experiment. In previous experiments carried out by our research group on aortic and kidney rings, an increase in cGMP was demonstrated through docking. In the study by Silveira (2019), the production of tissue cGMP induced by FOR811A was measured in aortic rings and revealed an increase in this nucleotide between the control and the group tested with FOR811A. Another important point is that this increase occurred even in the presence of ODQ, a selective inhibitor of GCs that was used to verify the participation of this enzyme in the relaxation induced by compounds that possibly act in this pathway (ZHAO et al., 2000). Sensitivity to inhibition of vascular relaxation by ODQ indicates a predominant heme mechanism for sGC activation, while resistance to ODQ suggests the possible presence of alternative vasorelaxing mechanisms. When incubating with ODQ, it was observed that there was no significant change in tissue cGMP values between the preparations devoid of this blocker and tested with FOR811A. This could suggest that the FOR811A may possibly act in a different way - such as direct activation of the channels for K+. Another behavior that can be estimated from the molecules is that they are activators of sGC, since, even with the oxidized enzyme, there was no change in the tissue expression of the cyclic nucleotide; a fact that would occur in the opposite way if the compounds were stimulators of sGC, resulting in a reduction of tissue expression of cGMP (PRIVIERO et al., 2005). In another study conducted by our group, the increase in cGMP in renal tissue was also visualized (ALVES, 2018), corroborating the results of Silveira (2019) and the results presented in our article, through docking.

● ALVES, N.T.Q. Renal effects of ruthenium complexes and their action in the protection of acute injury induced by ischemia and reperfusion. Thesis, Post-Graduation Program in Pharmacology, Federal University of Ceará, 2018. 109 p. [in Portuguese]

● PRIVIERO, F. B. M. et al. Mechanisms underlying relaxation of rabbit aorta by BAY 41-2272, a nitric oxide independent soluble guanylate cyclase activator. Clin. Exp. Pharmacol. Physiol., v. 32, n. 9, p. 728-734, 2005.

● SILVEIRA, J.A.M. Pharmacological characterization of the vasodilating activity of new ruthenium complexes containing imidazole derivatives. Thesis, Post-Graduation Program in Pharmacology, Federal University of Ceará, 2019. 147 p. [in Portuguese]

● ZHAO, Y. et al. Inhibition of soluble guanylate cyclase by ODQ. Biochemistry, v. 39, n. 35, p. 10848-10854, 2000.

In this way, we have performed changes for text improvement. therefore, the sentence “This anti-asthmatic effect can be explained by the activation of the enzyme soluble guanylate cyclase (sGC) in the cysteine portion, causing the release of NO by the metallodrug.” (lines 421-423) has been updated to “This anti-asthmatic effect could be explained by the activation of the enzyme sGC through its cysteine residue within the heme site, causing also the release of NO by the metallocompound, and further studies should be performed to confirm this hypothesis” (lines 453-456).

5) The paper needs to be proofread by a native English speaker to correct several inaccuracies in English.

Answer: You are right. As required, a professional did the corrections in the present version. 

Minor comments:

1) In the “Experimental Design and Treatments” the authors should define the meaning of “Sal” (page 4, line 107).

Answer: You are right. The sentence “…untreated controls (Ctl+Sal); control treated with FOR811A (Ctl+FOR); untreated asthmatic (Ast+Sal); asthmatic treated with FOR811A (Ast+FOR).” (lines 106-108) has been updated to “…untreated control receiving saline solution (Ctl+Sal); control treated with FOR811A (Ctl+FOR); untreated asthmatic receiving saline solution (Ast+Sal); asthmatic treated with FOR811A (Ast+FOR).” (lines 121-124).

2) Please rewrite Figure Legend 1 because is confused and unclear.

Answer: Right. The figure legend 1 “Fig. 1. Flat chemical structure of the ruthenium complex cis-[Ru(bpy)2(2-MIM)(NO)]3(PG6)3.” has been updated to “Fig. 1. Chemical structure of the ruthenium complex cis-[Ru(bpy)2(2-MIM)(NO)](PF6)3.”.

3) In the caption of Table 1 the authors state: “Values significantly different (p<0.05) by one-way ANOVA followed by Student–Newman–Keuls test compared to the Ctl+Sal group (*), and no difference compared to Ctl+Sal group (a).” In my opinion, the authors could avoid indicating the absence of statistically significant differences and remove “a”.

Answer: Right. The sentence “Values significantly different (p<0.05) by one-way ANOVA followed by Student–Newman–Keuls test compared to the Ctl+Sal group (*), and no difference compared to Ctl+Sal group (a).” (lines 309-311) has been updated to “Values significantly different (p<0.05) by one-way ANOVA followed by Student–Newman–Keuls test compared to the Ctl+Sal group (*), and no difference compared to Ctl+Sal group.” (lines 340-342) and the letter “a” was removed.

4) Figure 3 (legend): the authors should describe the figure more clearly.

Answer: Right. The sentence figure legend 3 “Fig. 3. Interaction of the FOR811A compound, a ruthenium oxide donor metallodrug with anti-asthmatic potential, on the Heme portion of the soluble guanylate cyclase (sGC) enzyme. Through the exploration of the drug-protein binding mechanisms by the molecular docking method, it was observed that FOR811A bound strongly to the distal portion of the Heme group of sGC, specifically in the residue Cys141 (a). When interacted with H-NOX, the FOR811A compound interacted with the hydrogen molecule through the cysteine at position 141 (Cys 141) of the 1.34 Å complex (b, c).” has been updated to “Fig. 3. Interaction of the FOR811A compound, a ruthenium nitric oxide donor metallocompound with anti-asthmatic potential, on the Heme portion of the soluble guanylate cyclase (sGC) enzyme. Through the exploration of the potential drug-protein binding mechanisms by the molecular docking method, it was observed that FOR811A bound strongly to the distal portion of the Heme group of sGC, specifically interacting with the residue Cys141 (a). In this study with H-NOX, FOR811A was shown to interact with the thiol portion of the cysteine residue at position 141 (Cys 141) (b, c).”.

Sincerely, 

The authors

---

## [Decision Letter · Decision Letter 1]

26 Feb 2021

Anti-asthmatic effect of nitric oxide metallo-donor FOR811A [cis-[Ru(bpy)2(2-MIM)(NO)]3+(PF6)3] in the respiratory mechanics of Swiss mice

PONE-D-20-30462R1

Dear Dr. Stefanie Waller,

We’re pleased to inform you that your manuscript has been judged scientifically suitable for publication and will be formally accepted for publication once it meets all outstanding technical requirements.

Kind regards,

Fabio Luigi Massimo Ricciardolo

Academic Editor

PLOS ONE

Additional Editor Comments (optional):

Reviewers' comments:

Reviewer's Responses to Questions

**Comments to the Author**

1. If the authors have adequately addressed your comments raised in a previous round of review and you feel that this manuscript is now acceptable for publication, you may indicate that here to bypass the “Comments to the Author” section, enter your conflict of interest statement in the “Confidential to Editor” section, and submit your "Accept" recommendation.

Reviewer #1: All comments have been addressed

Reviewer #2: All comments have been addressed

2. Is the manuscript technically sound, and do the data support the conclusions?

Reviewer #1: Yes

Reviewer #2: Yes

3. Has the statistical analysis been performed appropriately and rigorously? 

Reviewer #1: Yes

Reviewer #2: Yes

4. Have the authors made all data underlying the findings in their manuscript fully available?

Reviewer #1: Yes

Reviewer #2: Yes

5. Is the manuscript presented in an intelligible fashion and written in standard English?

Reviewer #1: Yes

Reviewer #2: Yes

6. Review Comments to the Author

Reviewer #1: The authors responded adequately to the points previously highlighted. No further changes would seem necessary

Reviewer #2: In my opinion, the authors amended the text and answered to the reviewer questions appropriately. I have no further concerns.

The paper is now considered to be suitable for publication in Plos One

7. PLOS authors have the option to publish the peer review history of their article (what does this mean?). If published, this will include your full peer review and any attached files.

Reviewer #1: No

Reviewer #2: No

---

## [Editor Report · Acceptance letter]

3 Mar 2021

PONE-D-20-30462R1 

Anti-asthmatic effect of nitric oxide metallo-donor FOR811A [*cis*-[Ru(bpy)_2_(2-MIM)(NO)](PF_6_)_3_] in the respiratory mechanics of Swiss mice 

Dear Dr. Waller:

I'm pleased to inform you that your manuscript has been deemed suitable for publication in PLOS ONE. Congratulations! Your manuscript is now with our production department. 

Kind regards, 

on behalf of

Professor Fabio Luigi Massimo Ricciardolo 

Academic Editor

PLOS ONE